# Intelligent Biosensors Promise Smarter Solutions in Food Safety 4.0

**DOI:** 10.3390/foods13020235

**Published:** 2024-01-11

**Authors:** Yuehua Chen, Yicheng Wang, Yiran Zhang, Xin Wang, Chen Zhang, Nan Cheng

**Affiliations:** 1School of Electrical and Information, Northeast Agricultural University, Harbin 150030, China; chyh@neau.edu.cn; 2School of Food Science, Northeast Agricultural University, Harbin 150030, China; wangycbtts@163.com; 3College of Food Science & Nutritional Engineering, China Agricultural University, Beijing 100083, China; sy20223061318@cau.edu.cn (Y.Z.); sy2223061315@cau.edu.cn (C.Z.)

**Keywords:** food safety risks, Internet of Things, smart packaging, Food Safety 4.0, digital technology, advanced functionality

## Abstract

Food safety is closely related to human health. However, the regulation and testing processes for food safety are intricate and resource-intensive. Therefore, it is necessary to address food safety risks using a combination of deep learning, the Internet of Things, smartphones, quick response codes, smart packaging, and other smart technologies. Intelligent designs that combine digital systems and advanced functionalities with biosensors hold great promise for revolutionizing current food safety practices. This review introduces the concept of Food Safety 4.0, and discusses the impact of intelligent biosensors, which offer attractive smarter solutions, including real-time monitoring, predictive analytics, enhanced traceability, and consumer empowerment, helping improve risk management and ensure the highest standards of food safety.

## 1. Introduction

Ensuring food safety has become increasingly challenging in today’s complex supply chains and global trade. Traditional food safety testing methods face several challenges, including long testing cycles, destructiveness, single-point testing, limited quantitative accuracy, poor reproducibility, unstable detection of large molecules, and the extensive training required for laboratory operators [1,2]. Food safety monitoring techniques and equipment are inadequate for meeting the demands of the rapidly evolving food market in the new era. Traditional food safety practices rely primarily on reactive measures to address problems. This approach results in delayed food safety risk monitoring, early warnings, and assessments, thereby limiting its effectiveness in reducing risk. Recently, the fourth industrial revolution (Industry 4.0) has emerged as a transformative force in the food industry, driven by technological advancements and an increasing demand for safer and more transparent food. This signifies the future of food safety practices, and has led to substantial changes.

Since the advent of the industrial revolution, the evolution of manufacturing has been denoted as Industry 1.0, a concept that has been further refined with the emergence of electrification and automation in Industry 2.0. The progression to Industry 3.0 is rooted in the widespread integration of electronics and information technology, and culminates in the substantial enhancement of the automation and control aspects of the manufacturing process [3]. Industry 4.0 embodies advanced production technologies, incorporating contemporary elements such as big data, artificial intelligence, the Internet of Things (IoT), machine learning, deep learning, cloud computing, system integration, and smart sensors. These technologies are instrumental in collecting, storing, analyzing, and utilizing extensive datasets within the agricultural and food industries [4,5]. The convergence of Industry 4.0 with agriculture is poised to catalyze the fourth agricultural revolution, exerting a profound influence on production-centric models, processes, and supply chains inherent in the agricultural sector. This synergy has given rise to the conceptualization of Agriculture 4.0 [6]. IoT and technologies allow access to agricultural information, such as; crop yield forecasting, pest and disease identification [7,8], automation of agricultural production through robots and autonomous systems to increase farm productivity [9], artificial intelligence for animal health detection, pest and disease identification, prediction of crop yields or weather conditions [10,11], and big data technologies that can provide farmers with intelligent agricultural recommendations [12]. Blockchain has traced the agricultural supply chain, enhanced the transparency and credibility of the supply chain, and improved consumer confidence [13]. These technological advancements are not merely methodological responses to challenges arising from the industrialized agricultural model; rather, they are concerted efforts to meet escalating technological demands associated with food safety issues. In recent years, the food industry has progressively embraced Industry 4.0, introducing several Industry 4.0-based terminologies (Table 1) that collectively steer the evolution of the food industry.

Based on our previous work [24,25], and inspired by the industrial changes caused by the information technology of Industry 4.0, and the impact of Agriculture 4.0 on food safety in the process of industrialized agriculture, we further developed the concept of “Food Safety 4.0”. Food Safety 4.0 advances traditional food safety practices by embracing a proactive data-driven approach that harnesses advanced digital technologies to predict, prevent, and swiftly address potential food safety risks. This represents a contemporary paradigm in food safety management, marking its evolution into a more sophisticated stage. Food Safety 4.0 prioritizes a proactive and predictive stance over reactive responses. Instead of relying solely on post-incident countermeasures and product recall, it integrates innovative technologies and data-driven strategies. These leverage the capabilities of digital technologies to facilitate continuous real-time monitoring across the entire food supply chain, fostering analytics-driven decision making. This innovative paradigm, Food Safety 4.0, aims to revolutionize the food industry by enhancing its resilience, efficiency, and capacity to provide consumers with safe and high-quality food. The foundational pillars of Food Safety 4.0 encompass real-time monitoring, predictive analytics, enhanced traceability, and consumer empowerment.

In the context of Food Safety 4.0, intelligent biosensors play a pivotal role in transforming conventional methodologies into proactive data-driven strategies. These biosensors facilitate early identification of potential hazards, including pathogens, contaminants, allergens, and quality parameters. This capability empowers organizations to preemptively recognize and address potential food safety risks, thus facilitating the continuous enhancement of safety standards. Intelligent biosensors contribute to the identification of contamination sources and quality issues by generating digital records of the food flow and conditions. This, in turn, increases traceability and transparency in the food industry. The applications of intelligent biosensors in the food industry are diverse and seamlessly integrated into all stages of food production, processing, distribution, and consumption. Through constant monitoring of the entire food supply chain, from production to consumption, these biosensors provide invaluable data. In turn, these data enable stakeholders to make well-informed decisions, ensuring both food safety and quality. Notably, their integration into personalized nutritional applications can address individual dietary needs, including allergies and sensitivity. This integration ensures that food is not only safer and more precisely targeted, but also affords consumers a secure and tailored food experience.

The main objective of this review was to explore the transformative potential of intelligent biosensors within the framework of Food Safety 4.0. Our examination delves into the purpose and scope of biosensor technology, comprehends the distinctive capabilities of intelligent biosensors, employs case studies to illustrate their real-world applications, evaluates the advantages and challenges linked to the incorporation of intelligent biosensor technology, and offers insight into the integration of these sensors into established food safety initiatives.

## 2. Intension and Extension Intelligent Biosensors

A biosensor is an analytical device comprising a biological element and signal converter. Biological elements, such as enzymes, antibodies, nucleic acids, or entire cells, generate a measurable signal when they interact with a target analyte, thus providing valuable information about its presence, concentration, or activity [9]. The sensor in the biosensor converts the biometric signal into a quantified electrical formula, which is critical for enabling biosensors to interact with smart technologies, and has become an important part of Food Safety 4.0 data-driven solutions.

Intelligent biosensors are innovative devices that combine biosensing technologies with digital systems and other advanced functionalities. The relationship between the various digital technologies is shown in Figure 1. Digital technologies enhance the capacity of biosensors to detect and analyze specific substances or biological processes with a high degree of sophistication. Concurrently, advanced functionalities, including smartphones, quick response (QR) codes, and smart packaging enable these sensors to detect and quantify specific substances but also execute a range of advanced functions such as real-time data analysis, processing, and communication. This multifaceted capability positions them as transformative tools with heightened effectiveness across various food safety applications.

Traditional biosensors typically require manual intervention for data retrieval and processing, which results in inefficiencies in swiftly changing the food safety landscape. In contrast, intelligent biosensors distinguish themselves from their traditional counterparts in terms of sophistication, connectivity, and autonomy. Notably, they offer real-time insights, facilitate proactive measures, and bolster decision-making capabilities in response to food safety risks. These distinctions arise from the integration of intelligent biosensors with digital systems, artificial intelligence, and other advanced technologies, which enables them to overcome the constraints inherent in traditional biosensors. The variances between intelligent and traditional biosensors are presented in Table 2.

Biosensors are increasingly being combined with digital technologies and advanced features, such as artificial intelligence and the Internet of Things, to enable smart functions with the assistance of smart devices. Ongoing data collection and secure storage provide detailed insights into the entire process of a product, from production to consumption. Thus, intelligent biosensors can transform food safety practices and contribute to the development of Food Safety 4.0 initiatives through a more efficient and proactive approach.

## 3. Applications of Intelligent Biosensors in Food Safety

### 3.1. Deep Learning-Based Intelligent Biosensors

Machine learning, sometimes called deep learning, is at the cutting-edge of artificial intelligence. There is a relationship between artificial intelligence, machine learning, and deep learning. This relationship can be understood as follows: artificial intelligence is the end, machine learning is the means, and deep learning is the best way to achieve this goal [26,27]. Owing to its strong ability to fit functions, deep learning has been widely researched as an effective machine-learning algorithm [28,29], with a large number of successful cases in the fields of medicine, agriculture, and hydrology [30,31,32].

Copious real-time data generated by intelligent biosensors pose challenges in data management, storage, and analysis, requiring robust data infrastructure and analytics capability. Deep learning has emerged as a potent method and tool for data analysis and processing and offers distinct advantages. First, it efficiently processes large datasets, which is particularly beneficial for deep learning-based biosensors that handle complex matrices or extensive sensor detection data. In contrast, traditional biosensors often link a single data feature to an indicator for sample concentration detection. Deep learning-based biosensors employing data visualization discern the intricate relationship between sample parameters and sensor signals [26]. Second, deep learning excels in extracting high-quality signals from noisy ones, facilitating the removal of contaminants from real samples and enhancing the performance of traditional biosensors [33]. Given the proficiency of deep learning in feature learning, various architectures such as convolutional neural networks (CNN), recurrent neural networks, deep neural networks (DNN), deep convolutional neural networks (DCNN), backpropagation neural networks (BPNN), long short-term memory networks, and transfer learning have been proposed. Among these, CNN, which are renowned for their advantages in image analysis, have become popular [34]. The core components of a CNN include a convolutional layer, pooling layer, convolutional kernel, activation function, and fully connected layer. The convolutional layer, which is crucial for image extraction, typically comprises multiple convolutional layers with kernels of varying sizes, which are strategically designed for feature extraction across the image. The fully connected layers serve as outputs and are adapted to different layers for model classification or regression tasks. CNN-assisted biosensors exhibit superior prediction accuracy for moderate-to-large datasets. For instance, a CNN-based surface-enhanced Raman scattering biosensor demonstrated exceptional capability by accurately identifying minute oligonucleotide damage on a gold grid substrate, which is a challenge for other techniques. This biosensor achieved a classification accuracy of 98% and confidence level exceeding 95% [35]. However, when processing fluorescent dot matrix images, challenges arise owing to variations in the light intensity in the testing environment, leading to potential false positives or false negatives.

Deep learning is instrumental in directly, accurately, and expeditiously enhancing biosensor readings, thereby advancing the qualitative identification of intricate overlapping signals, the quantitative prediction of trace analytes, and effectively addressing food safety challenges. The proposed CNN model architecture, trained through image processing, learns from diverse color solutions of the pesticide dichlorolyphenol (DCV) to predict the DCV concentration in the test solution. This approach achieved superior quantitative DCV detection with an accuracy of 97.6% and a detection range of 0–60 mM [36]. Applicable to on-site detection of contaminants in water and fruit juices, this method provides swift and quantitative insights into contamination. Hu et al. [37] devised a fluorescent biosensor utilizing a low-gradient magnetic field for highly sensitive detection of *Salmonella typhimurium* (Figure 2). Influenced by a magnetic field, the labeled magnetic bacteria transform from a three-bit spatial distribution to a two-dimensional planar distribution. Deep learning employing a faster region-based CNN (R−CNN) identifies fluorescence spots on bacterial cells for the identification of target bacteria. The model was built using the complete fluorescence image dataset and harnessed numerous convolutional layers for effective feature extraction, achieving quantitative detection of *Salmonella typhimurium* in the range of 6.9 × 10^1^–1.1 × 10^3^ colony forming units (CFU)/mL, with a lower limit of detection of 55 CFU/mL in 2.5 h. This method allows for the simultaneous detection of multiple foodborne pathogens by utilizing distinct antibodies and fluorescent materials of various colors. Deep learning methodologies open novel avenues for predicting bacteria in food products. Nehal et al. [38] simulated an optical biosensor based on a CNN deep-learning model of photonic crystals, generating training data for predicting samples with features, but no labels. This model explicitly predicted the presence of *E. coli* in water, achieving a detection accuracy of 95% based on the output spectrum and identification of training data. Smart sensors play a pivotal role in field testing and ensuring the accuracy and reliability of biosensor readings requires regular calibration and maintenance. This entails adhering to proper maintenance procedures and engaging qualified personnel to maintain an optimal biosensor performance.

As heightened concern for food safety has gained prominence, there is a growing demand for real-time monitoring to promptly detect issues and effectively take corrective actions in response to food safety risks. Jia et al. [39] pioneered the development of an innovative colorimetric sensor array designed for real-time monitoring of beef freshness. Sensor tag videos were captured in a color controller lightbox using an a smartphone to obtain image sets. These images were then randomly divided into training and test sets and imported into convolutional neural networks (CNNs) to train the food freshness classification system. The results demonstrated that all four CNN architectures successfully distinguished freshness with an accuracy exceeding 96%, thereby enhancing the accuracy of biosensor recognition. In another study, Ma et al. [40] used UiO-66-Br (selected for its highest binding energy) on an ice-templated chitosan substrate (ice-templated dye @UiO) to construct sensor arrays (Figure 3A). They established a sensitive, non-destructive product platform, enabling customers to easily monitor shrimp freshness in real time using the Wide-Slice Residual Network 50, which achieved an impressive accuracy of up to 99.94%. Four state-of-the-art DCNN models were trained with 31,584 labeled images, and 13,537 images were reserved for testing. Furthermore, deep learning systems integrated into the computer vision can be employed to monitor food production. Iheonye et al. [41] successfully employed computer vision, combined with deep learning systems, to address the issue of the fluidized bed food drying process, which poses challenges in capturing food images.

Intelligent biosensors that leverage deep learning have found effective applications in food classification, thereby advancing food safety management practices and fortifying consumer rights. The method for discriminating the age of Pu’er tea using a voltammetric electronic tongue is shown (Figure 3B) [42]. One-dimensional CNN (1−D CNN) was employed for automatic feature extraction and classification, with transfer learning incorporated to reduce the model training complexity and enhance the generalization capacity of CNN for electronic tongue applications. The efficacy of this method was compared with that of traditional machine learning techniques, including support vector machines, back-propagation neural networks, and limiting learning machines. These results highlight the superior performance of the proposed model for the Pu’er tea classification. In a separate study, Yang et al. [43] developed a transfer learning framework based on a BPNN for classifying various wine types. This framework refines only the output layer, demonstrating robust performance in recognizing diverse wine types based on their characteristics. This approach has been extended to the classification of different Chinese white wines, thereby mitigating the costs associated with model training.

In conclusion, the incorporation of deep learning into predictive analytics, particularly in biosensor applications, provides novel opportunities for precise and efficient data analysis. The resilience of deep learning architectures and their ability to address various data challenges establish them as essential instruments for enhancing the capabilities of predictive analytics.

### 3.2. IoT-Based Intelligent Biosensors

IoT refers to a network that combines the Internet, traditional telecommunications networks, and other information carriers, facilitating interconnection and interoperability among various physical devices, each of which is independently addressable. Essentially, it constitutes a network expanding through the Internet to realize the interconnectivity of devices, people, and things globally and at all times. IoT encompasses intelligent objects equipped with sensors, networking, and processing technologies that are harmoniously integrated to deliver intelligent services, presenting unprecedented opportunities for addressing large-scale challenges in application areas such as precision agriculture, environmental monitoring, smart manufacturing, smart cities, and smart health [44,45].

Sensors within the IoT framework can precisely detect factors affecting food quality, monitor perishable items, identify food adulteration, and gather real-time product information. The collected data are shared across a network of interconnected smart devices to enable early warnings and enhance the reliability of food safety tracking and tracing [46,47]. Seo et al. [48] presented a compact immune-sensing system for monitoring foodborne pathogens managed by a dedicated application. This application uploads the results to a server via a wireless network for public access, such as the presence of pathogens such as *Vibrio parahaemolyticus.* This approach not only addresses the challenge of using diverse mobile devices in the field, but also exemplifies pathogen monitoring through IoT. To address the diminishing activity of immobilized enzymes over time in certain biosensors, an intelligent portable biosensor was developed (Figure 4A) [49]. This biosensor comprises an enzyme-based three-electrode electrochemical unit, signal processing and wireless data transmission unit, iOS platform-based application, decision-making unit, and unit for sharing results via an IoT cloud server. The nitrate determination results were transferred to a server for storage in the cloud, facilitating biosensor commercialization and remote monitoring. However, stakeholders must address concerns about data security and privacy arising from digital integration, emphasizing the need to safeguard sensitive information related to food, consumers, and operations.

The IoT can leverage biosensors integrated with wireless transmission technology and wearable devices. In a livestock health application, an implantable biosensor system was employed for the real-time measurement of the subcutaneous temperature at the base of a cow’s ear with the aim of mitigating the risk of disease in cows (Figure 4B) [50]. This real-time monitoring system comprises an implantable biosensor, a wearable radio frequency identification (RFID) scanner, and a long−range (LoRa) hub. LoRa radio technology facilitates the transmission of signals from the RFID scanner to the LoRa hub, whereas WiFi technology is employed to transmit data from the hub to the cloud server. Compared to Bluetooth devices, LoRa wireless technology demonstrates lower energy consumption and a broader transmission range. Wearable sensors are portable and are ideal platforms for continuous real-time health monitoring. They can detect substances such as lactic acid or alcohol in sweat or tissue fluids, thus playing a significant role in health and medicine [51]. The real-time and portable nature of wearable biosensors has fostered a growing need for their application in on-site food safety detection. Ma et al. [52] developed a flexible electrochemical biosensor and supercapacitor using smart ink and screen-printing technology for efficient detection of hydrogen peroxide and ascorbic acid (AA). This technology is effective for monitoring juice and VC tablet samples for hydrogen peroxide and AA. Mishra et al. [53] proposed a glove-based, flexible electrochemical biosensor. The biosensing system on the index finger can be used for the on-site detection of organophosphate nerve agents by swiping, and the results can be transmitted in real time to a smartphone device via wireless data.

The integration of the IoT into traceability technology enhances food safety monitoring, promotes information transparency, and contributes to rebuilding trust between consumers and the market [54]. Food enterprises should optimize the use of the IoT for informed decision-making and establish a transparent digital food supply chain. This boosts consumer confidence by providing real-time ingredient tracking from the field to the dinner table, offering consumers with comprehensive information about their purchased ingredients. However, to ensure the seamless operation of intelligent biosensors integrated into existing food safety systems and supply chain processes, adjustments and compatibility assessments may be necessary. The initial investment required to acquire and integrate such technologies may pose a significant challenge for small food businesses with limited resources, potentially acting as a barrier to market entry.

### 3.3. Smartphone-Based Intelligent Biosensors

Smartphones have become ubiquitous in modern life. Their applications extend to mere communication tools, with functional boundaries continually expanding owing to increased computing power, 5G connectivity, and the integration of multiple sensors [55]. For example, smartphone applications can be used to assist with tsunami evacuation, accelerate forest inventories, and determine sodium intake [56,57,58,59,60,61,62,63,64,65,66,67,68].

A portable and user-friendly smartphone-based biosensor for detecting ochratoxin A (OTA) in wine and instant coffee is proposed (Figure 5A) [59]. The smartphone camera serves as a light detector, and a low-cost disposable analytical column contains lateral flow immunoassay test strips and necessary reagents, allowing non-professional operators to perform analyses as needed through simple manual operations. The biosensor generated a signal that was inversely proportional to the amount of OTA in the sample, enabling reliable quantification in compliance with the current regulations for rapid and confirmatory product identification. Wang et al. [60] developed a smartphone-based fluorescent biosensor for the rapid detection of *S. typhimurium.* A smartphone application using an inter-frame difference algorithm enables the online counting of fluorescent spots to determine the number of target bacteria. By leveraging smartphones’ analytical and graphical processing capabilities, optical signals, including red, green, blue [61], and hue-saturated luminance [62], can be analyzed in real time, facilitating the detection and evaluation of targeted food products to enhance food safety. Smartphone-based biosensors excel in image acquisition, but face the challenge of ambient light interference, which is typically addressed by adopting 3D printing technology. The sensor incorporated a 3D-printed cassette as a smartphone signal adapter, and the reusable sensing paper holder was 3D-printed to ensure standardized, fast, and robust smartphone detection of all three biosensing responses [63]. Montali et al. [64] developed a smartphone-based collapsible biosensor to detect acetylcholine inhibitors (Figure 5B). A small 3D-printed black box minimizes the influence of ambient light and features a slot for insertion into the holder and an adapter for connection to the smartphone as a portable light detector. While intelligent biosensors are progressing towards greater portability and ease of handling, their successful implementation necessitates adequate training for those interacting with the technology.

Currently, smartphone-based biosensors predominantly target food safety detection, whereas bionic sensors such as electronic noses and tongues are emerging as potential tools for food safety assessment [65]. Electronic noses identify volatile compounds in food to assess their quality, and electronic tongues analyze their chemical properties. Integrating smartphone-based biosensors with bionic sensors can enhance the overall identification capabilities of the sensing system [66]. Wei et al. [67] devised a smartphone-controlled electronic nose-sensing array comprising 12 sensors to collect taste and aftertaste information (Figure 6). These data were then transmitted to a network platform via IEEE 802.11 (WiFi), facilitating on-site detection of yellow wine of various ages. In addition, IoT has the potential to transform smartphone-processed results from specialized tests into real-time and rapid tests. This transformation not only aids in food safety regulation but also empowers each consumer to become a guardian of Food Safety 4.0 through information sharing.

Overall, smartphone-based biosensors, coupled with advancements in bionic sensors and IoT integration, have significant potential for improving food safety assessments, fostering consumer empowerment, and facilitating the realization of Food Safety 4.0. It is essential to provide adequate training for users interacting with these technologies to ensure their successful implementation in practical scenarios.

### 3.4. QR Codes-Based Intelligent Biosensors

QR codes serve as a link between physical products and digital information, and are deemed advantageous tools for augmenting consumer empowerment. Consumers promptly access a wealth of information through a smartphone scan of QR codes, which enables them to evaluate the authenticity and safety of a product [68,69,70,71].

QR codes serve not only as a conduit for transmitting fundamental product information but are also a subject of ongoing research for direct involvement in food safety inspections. Yuan et al. [72] proposed the integration of paper-based microfluidics with QR codes to enable optical scanning using a smartphone. The chromaticity of QR codes has been harnessed to facilitate reliable reading. This was achieved by strategically immobilizing the target analyte antibodies in the decoding region of the QR code. In addition, gold nanorods labeled with antibody markers served as the signal elements within the system. Signal enhancement was attained through the application of pyrocatechol, which reduced the nanosilver to metallic silver on the surface of gold nanorods. This synergistic amalgamation of techniques ensures a legible and distinguishable QR code, enhancing its functionality as a dependable data carrier, even in the presence of potential errors or distortions.

In conclusion, QR codes have emerged not only as effective and versatile tools for intelligent biosensors, but also as significant contributors to consumer empowerment. Their capability to digitize and seamlessly integrate information empowers consumers by providing instant access to comprehensive product details and facilitating informed decision making. Furthermore, the potential for customization and interactivity enhances consumer experience, allowing active engagement with brands and the acquisition of additional resources tailored to individual preferences and needs. Ongoing research continues to explore the transformative potential of QR codes to revolutionize the field of biosensors. This exploration creates a path toward smarter, more accurate, and user-centric diagnostic solutions that empower consumers to manage their health and well-being. This paves the way for the food industry to thrive in the era of Food Safety 4.0.

### 3.5. Smart Packaging with Intelligent Biosensors

Smart packaging involves the integration of advanced technologies into packaging materials to enhance the product quality, safety, and functionality. With the integration of intelligent biosensors with advanced capabilities, smart packaging systems can revolutionize the monitoring, management, and overall experience of products across the supply chain. This transformation represents an effective approach to survey the condition of packaged foods by mitigating food waste and the incidence of foodborne illnesses.

The sensitivity, stability, and efficiency of sensors in smart biosensing packaging heavily rely on the design of the underlying materials [73]. For instance, the synthesis of metal-organic frameworks on the packaging material itself can be employed to manufacture nano-composite film, enhancing the structural performance of polymers while simultaneously acting as analyte receptors or signal transducers [74]. Song et al. [75] fabricated a Fe(II)-based metal-organic framework as a precursor for the detection of oxytetracycline (OTC) using iron oxide and mesoporous carbon materials. This material exhibited high selectivity, applicability, repeatability, stability, and regenerability, which demonstrated the ability to detect OTC in milk at concentrations as low as 0.027 pg/mL.

Polymeric substrates, including biodegradable polymers, such as cellulose, starch, and chitosan, along with conventional plastics, such as polystyrene and polyethylene, play a pivotal role as fundamental materials in smart food packaging, primarily because of their cost-effectiveness and tunability [76]. Prasad et al. [77] designed a packaging tray and a reagent-infused membrane to create a laboratory environment for packaging that can be universally paired with various pathogen sensors. This system allowed for the detection of target pathogens at 10^3^ CFU/g in packaged whole chicken samples (Figure 7A). Barandun et al. [78] incorporated a “zero-cost” printed electrical gas sensor within food packaging, which allows for the authentic detection of food freshness (Figure 7B). These sensors utilize the natural moisture-absorbing characteristics of cellulose in paper and measure the impedance to monitor the presence of water-soluble gases near the paper, such as NH3, which is a characteristic gas indicating food decay.

The emergence of novel materials has facilitated advanced functionalities, including analyte reception, signal transduction, and creation of laboratory-like conditions within packaging. Ongoing progress in materials science, sensor technology, and data analytics holds promise for further increasing the capability and efficiency of smart packaging. The integration of comprehensive solutions that incorporate artificial intelligence and machine learning algorithms may unlock opportunities for predictive analysis, preventive maintenance, and adaptive packaging designs.

## 4. Conclusions and Future Prospects

As the food industry enters the era of “Food Safety 4.0”, the integration of intelligent biosensors has emerged as a game-changing solution to food safety challenges. These advanced sensors offer advantages that promise to revolutionize food safety practices and enhance consumer protection.

Intelligent biosensors play a crucial role in early pathogen detection, and are often applied to monitor the freshness of foods such as meat, shrimp, and fruits, ensuring that foodborne illnesses caused by contamination or spoilage can be identified before they reach the consumer, preventing potential hazards from reaching the consumer. The traceability and transparency provided by intelligent biosensors combined with IoT enables the monitoring and tracking of every type of food product. For example, for cold chain food imported from overseas or food that needs to be exported or transferred through the information tracing mechanism, the responsible parties in each link of food production and distribution can be clarified to effectively prevent various food safety risks, and the integration of intelligent biosensors with data analytics and artificial intelligence allows for the prediction and real-time monitoring of the levels of harmful substances generated during animal breeding and product processing, enabling stakeholders to adopt the integration of intelligent biosensors with data analytics and artificial intelligence to predict and monitor the levels of hazardous substances generated during animal breeding, product processing, and other processes, enabling stakeholders to adopt proactive food safety management strategies, take preventive measures, and optimize food safety practices. Two essential factors contribute to the optimal realization of the benefits associated with intelligent biosensors: first, the need to persuade stakeholders about the advantages of adopting these biosensors and overcoming any resistance to change, and second, fostering collaboration among stakeholders, securing regulatory support, providing adequate training, and adopting a comprehensive implementation approach.

As the field of intelligent biosensors continues to develop, their landscape of intelligent biosensors is evolving with the integration of new materials, forms, and technologies using more portable, advanced, and efficient solutions for food safety. Wearable biosensors facilitate the convenient, personalized, and continuous monitoring of individual health parameters, food allergens, and contaminant exposure. Nanotechnology-based biosensors, offering possibilities for ultrasensitive pathogen detection and real-time surveillance, emerge as crucial tools for identifying low-level contaminants or pathogens in complex matrices. Quantum-based biosensors, including quantum dots and quantum resonance sensors, are promising for achieving analyte detection with heightened sensitivity and precision. Intelligent biosensors are expected to exhibit enhanced multiplexing capabilities, enabling simultaneous detection of multiple analytes. Furthermore, the trajectory of intelligent biosensor evolution is poised towards the creation of biodegradable and environmentally friendly variants, addressing concerns related to e-waste and aligning with the escalating emphasis on eco-friendly technologies. In addition to food safety, smart biosensors are applicable to environmental sensing, contributing to the monitoring of water quality, air quality, and soil conditions. Collectively, these diverse applications contribute to a more comprehensive understanding of potential risks to food safety.

The convergence of biosensors with innovative technologies, such as artificial intelligence and IoT, has broadened their societal impact, contributing to economic growth, serving the population, and generating societal benefits. This intersection introduces social challenges, including ethical dilemmas and privacy breaches. Increasing reliance on these technologies tends to foster dependence, eliciting anxiety and panic when issues arise and potentially diminishing people’s subjectivity in navigating an objective world. The inevitable challenge of large-scale data privacy breaches arises with smart technology, which enables extensive data collection and creates opportunities for malicious intent. Moreover, if smart biosensors cause irreversible harm to society, then the ethical dimension of responsibility warrants further consideration. Current research on, and development of intelligent biosensors requires a meticulous examination of potential ethical concerns, emphasizing a development paradigm rooted in meeting human needs without compromising ecological sustainability, the moral and ethical fabric of society, or the practical interests of the public.

Intelligent biosensors constitute a potent instrument for realizing the objectives of Food Safety 4.0. The food industry can establish a secure, transparent, and robust food supply chain by incorporating cutting-edge sensors into a holistic food safety framework. With ongoing technological advancements and strategic implementations, intelligent biosensors have been poised to shape the trajectory of future food safety management, rendering them more intelligent, proactive, and adept at safeguarding public health and bolstering consumer confidence.

## Figures and Tables

**Figure 1 foods-13-00235-f001:**
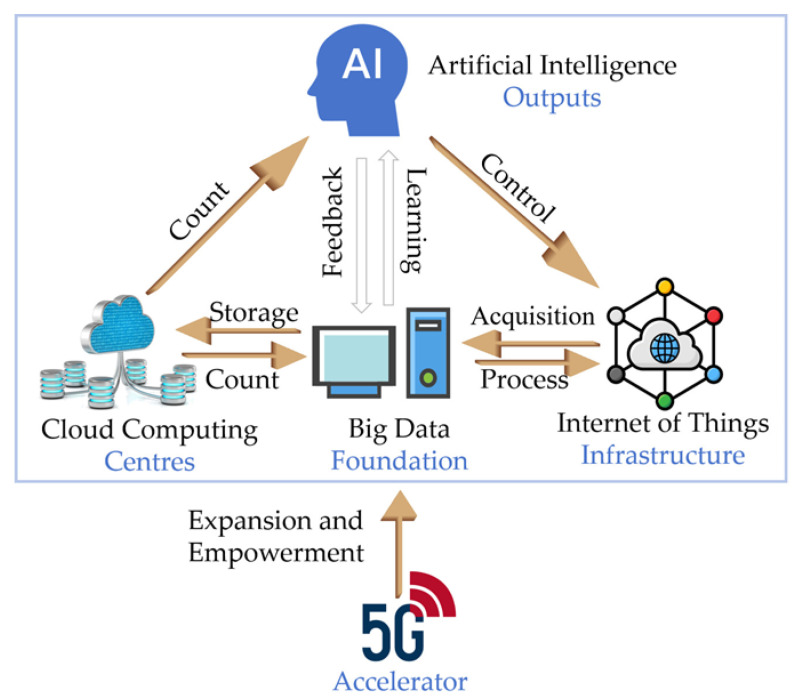
Relationship of various digital.

**Figure 2 foods-13-00235-f002:**
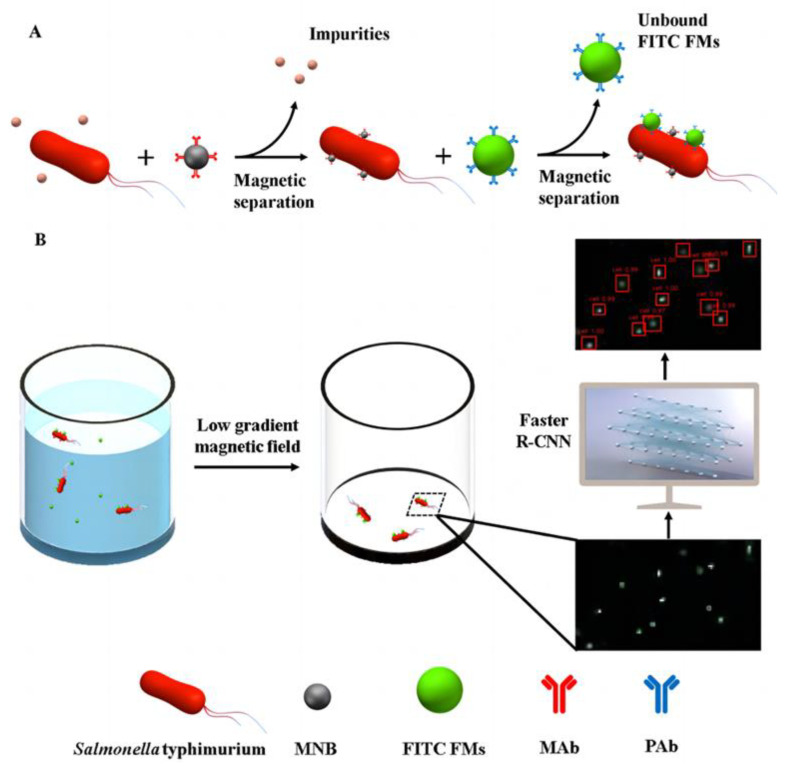
Schematic of this fluorescent biosensor. (**A**) Formation of fluorescent bacteria in the fluorescent biosensor. (**B**) Recognition and detection of fluorescent bacteria using a low-gradient magnetic field and deep learning via faster R-CNN [37].

**Figure 3 foods-13-00235-f003:**
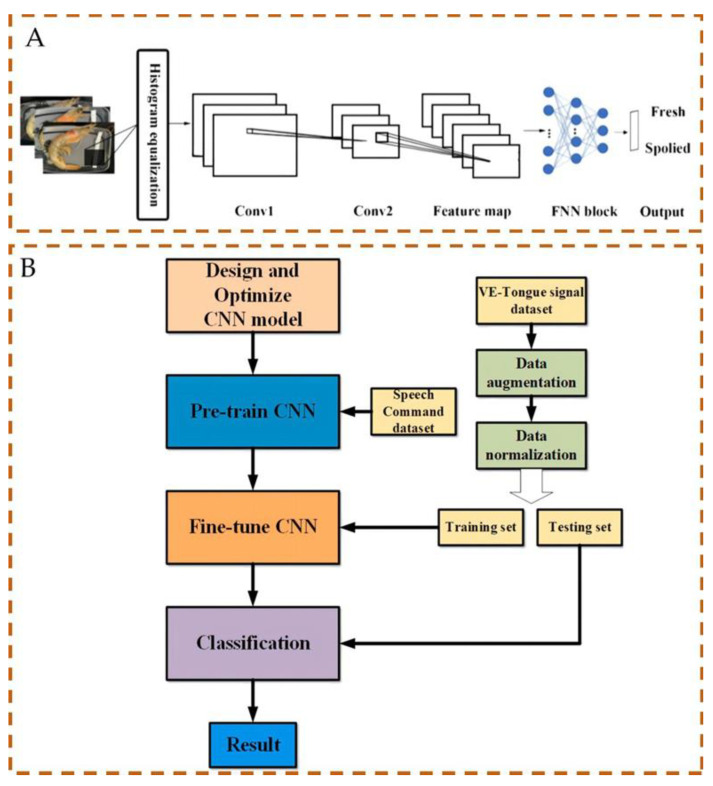
Two deep learning models. (**A**) The DCNN models were used for shrimp freshness classification, adapted with permission from [40] Copyright 2021, American Chemical Society. (**B**) Optimization structure of the 1-D CNN based on transfer learning, adapted with permission from [42] Copyright 2023, Elsevier.

**Figure 4 foods-13-00235-f004:**
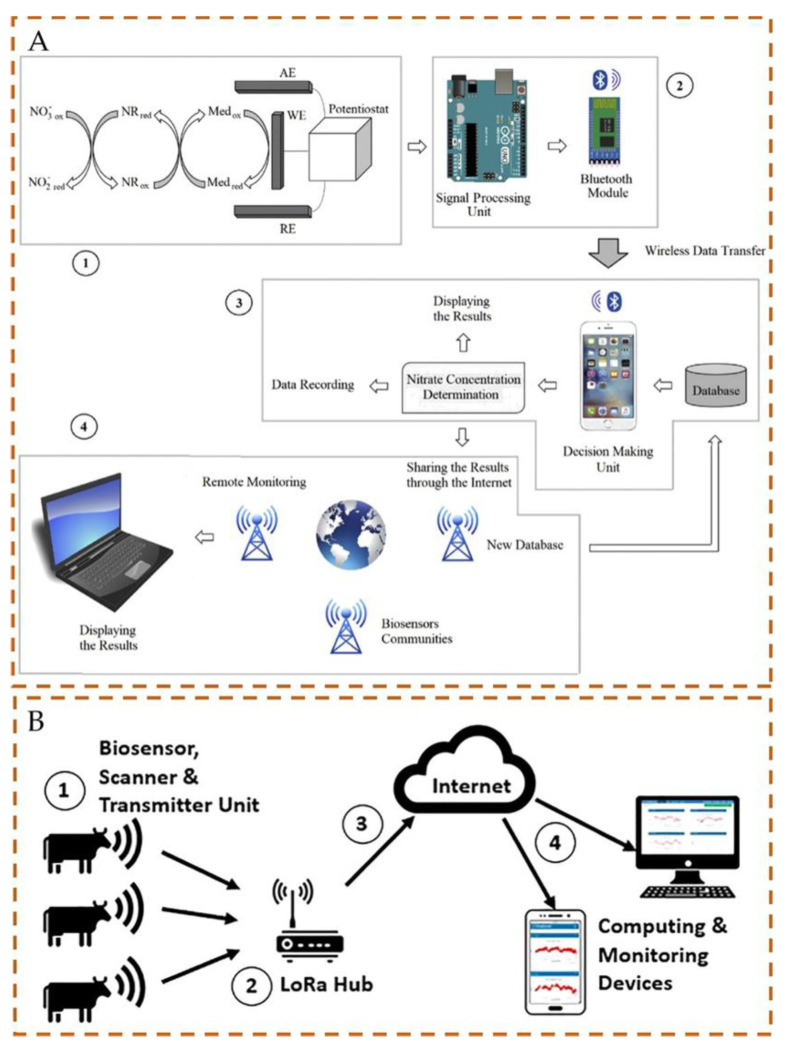
Schematic of IOT−based intelligent biosensor. (**A**) A schematic view of the proposed intelligent biosensor: ① Electrochemical nitrate biosensing unit; ② Signal processing unit and wireless data transfer; ③ Decision making unit; ④ Sharing the results through an Internet of Things-based cloud server, adapted with permission from [49], copyright 2023, Elsevier (**B**) A schematic of the method tested: ① the wearable RFID scanner reads the temperature data sent from the implanted biosensor; ② the transmitted temperature data sent from each scanner are collected by the central LoRa Hub; ③ a single temperature data string formed by the hub is sent to the cloud server (at fixed intervals); ④ the temperature data are plotted (and can thus be visualized by any personal computing devices), adapted with permission from [50], copyright 2023, Elsevier.

**Figure 5 foods-13-00235-f005:**
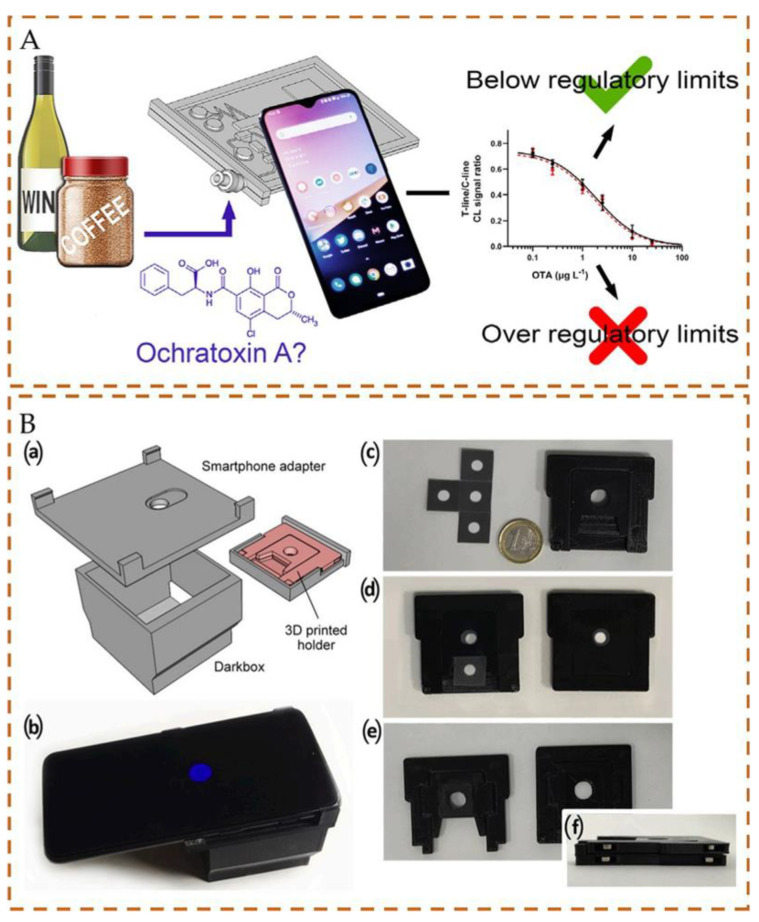
Schematic of a smartphone−based biosensor. (**A**) Schematic diagram of a smartphone-based portable biosensor for the quantitative detection of OTA in wine and instant coffee samples. The rightmost curves in the figure represent the calibration curves obtained in wine and instant coffee matrices (analysed in triplicate for each standard solution), with the black dots for white wine and the red dots for instant coffee ), adapted with permission from [59], copyright 2023, Elsevier.(**B**) Schematic diagram of the proposed smart biosensor: (**a**) Schematic drawing of the device. (**b**) Device connected to the One Plus 6 smartphone. (**c**) Unfolded paper-based biosensor and 3D printed holder. (**d**) The two parts compose the 3D printed holder; each part contains a 5 mm hole to enable the addition of luminol solution and acquisition of the chemiluminescence signal with the smartphone. (**e**) 3D printed holder (top and bottom view) housing the paper-based biosensor. (**f**) Detail of the assembled holder showing the N52 grade neodymium magnets designed to keep the paper biosensor folded, adapted with permission from [64], copyright 2023, Elsevier.

**Figure 6 foods-13-00235-f006:**
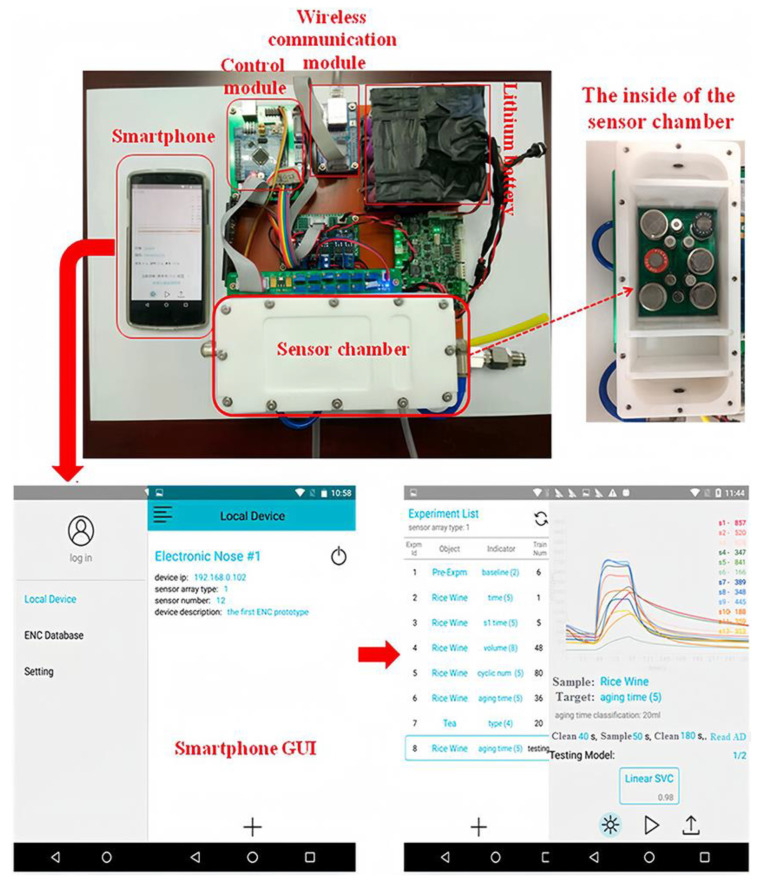
The set-up of the smartphone electronic nose [67].

**Figure 7 foods-13-00235-f007:**
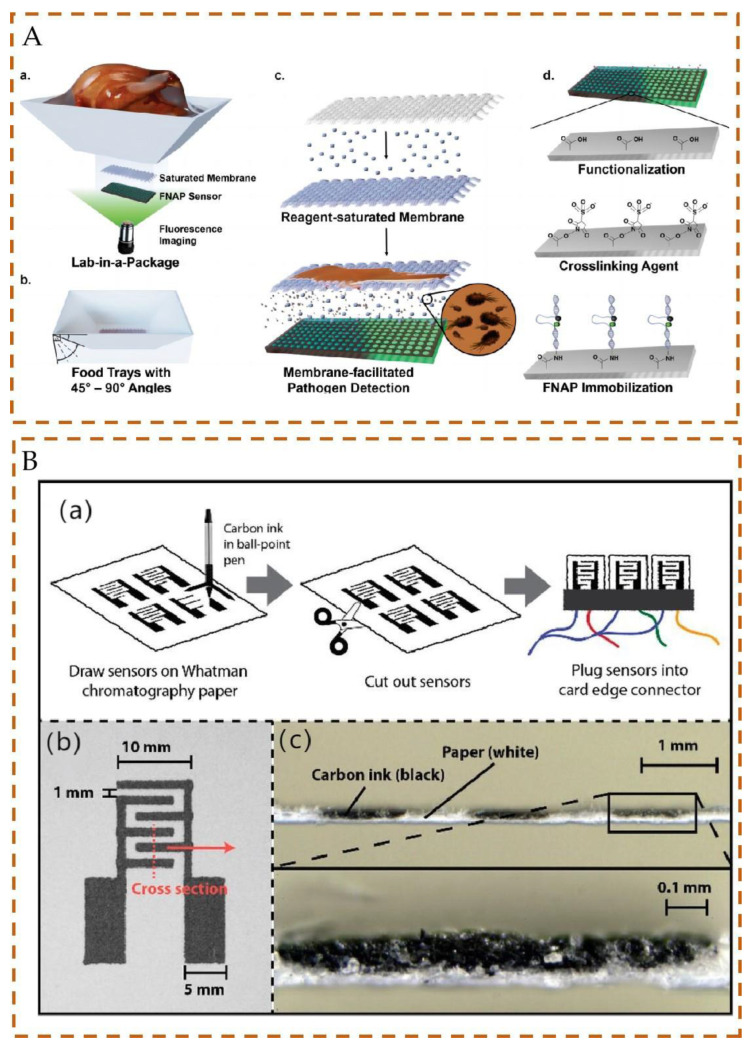
Schematic of smart packaging with intelligent biosensors (**A**) Schematic illustration of the Lab-in-a-Package platform: (**a**) Complete Lab-in-a-Package in situ detection platform with inclined packaging tray, reagent-saturated membrane, and sensor incorporation shown for ready-to-eat chicken products. Imaging procedure involving fluorescence scanning is also shown. (**b**) Inclined food packaging trays with angles ranging from 45° to 90° to optimize test sample localization. (**c**) Depiction of membrane saturation with reagent components, diffusion of buffer components and target analyte to sensor surface, and fouling prevention. (**d**) Fluorescent nucleic acid probe sensor development with corresponding material surface and biochemical modifications, adapted with permission from [77], copyright 2023, Advanced Materials. (**B**) Schematic diagram of an electrical gas sensor for detecting food freshness: (**a**) Fabrication of paper-based electrical gas sensors (PEGS). Carbon electrodes are printed on Whatman Chromatography 1 cellulose paper with a ballpoint pen and cutter plotter, allowing rapid prototyping in the desired geometry. Once printed, the sensors are cut and placed inside a card-edge connector for characterization. (**b**) Top view of a single PEGS consisting of two electrodes with three fingers and a spacing of 1 mm between each finger. (**c**) Cross-sectional view of a PEGS across three fingers (red dashed line in panel (**b**)). Carbon ink (black) partially penetrates paper (white), adapted with permission from [78], copyright 2019, American Chemical Society.

**Table 1 foods-13-00235-t001:** Industry 4.0 based terminology and purpose in the food sector.

Terminology	Purpose	References
Food quality 4.0	To ensure high food quality, save time and labor, and increase efficiency in the food industry	[14]
Food Logistics 4.0	To minimize resources and waste and effectively manage the transport of food from farm to fork while meeting consumer demand	[15]
Food Flavor Analysis 4.0	To achieve rapid detection of food additives, quality, and authenticity in food, and to accurately predict the flavor of unknown food samples	[16]
Food Traceability 4.0	To ensure the authenticity, safety, and high quality of food	[17]
Food processing 4.0	To improve the quality and safety of processed foods, reduce production costs and time, save energy and resources, and reduce food loss and waste	[18]
Agri-food 4.0	To support better supply chain decision-making processes	[19]
Sustainable Supply Chain 4.0	To organize a closed-loop product life cycle	[20]
Meat 4.0	To advance meat processing, preservation, quality, safety, and authenticity analysis techniques	[21]
Birth of Dairy 4.0	For more automation and optimization in the dairy industry	[22]
Packaging 4.0	For decentralized data collection in the supply chain, in-store and post-purchase phases, leading to consistent lifecycle monitoring	[23]

**Table 2 foods-13-00235-t002:** Differences between intelligent biosensors and traditional biosensors.

Features	Traditional Biosensors	Intelligent Biosensors
Multiplexing and Specificity	Usually, they only detect a single analyte	Multiple substances can be detected simultaneously and higher sensitivity and specificity can be achieved
Data analysis and connection	The signals generated usually require external equipment for manual interpretation or processing	Autonomous analysis by intelligent devices and connectivity to digital systems and networks, enabling seamless integration and real-time data transfer
Real-time Monitoring and Alerts	Requires manual activation and data retrieval, which may require more time to collect and analyze data, resulting in delays in detecting and addressing problems	Continuous real-time monitoring of specific substances or conditions and triggering of alarms or notifications when specific thresholds are exceeded
Decision-making	Usually provides passive data, detects and reports on specific substances as prompted	Autonomous decision-making through continuous monitoring and real-time analysis of data
Remote Accessibility	/	Remote access and monitoring via web interface or mobile app
Predictive Analytics	/	Intelligent biosensors can be integrated with AI and data analytics, enabling them to analyze complex data patterns and make predictions

## Data Availability

Data is contained within the article.

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
