# Peer review of "Intelligent Biosensors Promise Smarter Solutions in Food Safety 4.0"

_foods, 2024, doi:10.3390/foods13020235_

Round 1

Reviewer 1 Report

Comments and Suggestions for Authors

The manuscript entitled "Intelligent Biosensors Promise Smarter Solutions in Food Safety 4.0" aims to explore and emphasize the critical role of intelligent technologies, particularly intelligent biosensors, in addressing the intricate and resource-intensive challenges associated with food safety. The manuscript aims to introduce the concept of Food Safety 4.0, which involves the integration of digital systems and advanced functionalities with traditional biosensors. Through this integration, the manuscript seeks to showcase how intelligent design can revolutionize current practices in the field of food safety. The primary focus is on elucidating how intelligent biosensors offer attractive and smarter solutions, such as real-time monitoring, predictive analytics, enhanced traceability, and consumer empowerment. Ultimately, the manuscript aims to contribute to the improvement of risk management in the realm of food safety, ensuring the highest standards through the application of intelligent technologies. This topic is highly important nowadays. The manuscript is very interesting and fits well with the scope of the Journal. It is well-prepared in general. Still, some issues need to be addressed. My specific comments are given below.

While discussing the benefits of intelligent biosensors, the abstract briefly touches upon the challenges associated with food safety regulation and testing processes. A more comprehensive overview of these challenges could strengthen the manuscript. Also, the abstract mentions the implementation of "intelligent technologies" without specifying the types of technologies involved.

The introduction is informative and very well written.

Figure 1 is too small, and the visibility is poor. It should be adjusted.

Table 2 should contain border lines in order to be readable well. In the present form, it is hard to get the information.

Figure 2 should be removed.

Figures 4b, 4c, 5b, 5c and 7a are also poorly visible.  

The final paragraph, Conclusions and Future Prospects, should be written with more details. Consider the prospect of customizable biosensors tailored to specific food safety challenges. Discuss the potential for adaptive technologies that can evolve to address emerging risks and changing regulatory requirements. Address potential ethical considerations and societal implications associated with the increasing integration of intelligent biosensors. Consider the importance of responsible innovation and the need to balance technological progress with ethical considerations.

Comments on the Quality of English Language

Minor changes are required. 

Reviewer 2 Report

Comments and Suggestions for Authors

I revised the manuscript entitled “Intelligent Biosensors Promise Smarter Solutions in Food Safety 4.0” which deals with the concept of “Food Safety 4.0” with the focus being put on intelligent biosensors. I think this manuscript is well organized and can be of interest to many readers and researchers. Only a few suggestions can be made as follows:

-The last sentence of the abstract should be revised.

-The keywords list should be revised to avoid repeating words that are already present in the title

-L28: “Business” can be replaced by “agriculture and the food industry”

-L36: please replace “through” with “while”

- L47: In fact the concept of “Food Safety 4.0” has already been introduced in Chapter 4 of the book: Food Industry 4.0: Emerging Trends and Technologies in Sustainable Food Production and Consumption

-A short reminder of previous industrial revolutions (i.e., Industry 1.0, Industry 2.0, and Industry 3.0) could have been of relevance.   

-L83: I think the sentence “the sensors in a biosensor convert...” is not appropriate, please revise it

-Table 2: Decision Making: the word “Autonomous” can be deleted from the first column and added to the third column.

-L119: traditional biosensors are not combined with digital technologies, etc. Please rephrase. This could be “Biosensors are increasingly becoming combing… “.

-Can the resolution of Figure 3 (especially Part C) be improved?

-It is recommended to improve the resolution of Part A Figure 4, and Part C Figure 5 as well.

-L438: Please replace “field” with “era”.

-I think it is better to replace “intelligent biosensors” with “smart biosensors” throughout the whole manuscript, as smart sensors or biosensors are more used in the literature. Otherwise, the authors are invited to show the difference, if any, between smart and intelligent.

L445: the “portability” of biosensors could be envisioned as a future trend besides nanotechnology

Comments on the Quality of English Language

English is good and understandable

Reviewer 3 Report

Comments and Suggestions for Authors

The manuscript entitled "Intelligent Biosensors Promise Smarter Solutions in Food Safety 4.0" is original and well defined. Presented manuscript fits the journals scope in presentet form. References are up to date and the implementation of intelligent technologies is imperative to effectively address food safety risks and to protect consumers from food safety risk.

Round 2

Reviewer 1 Report

Comments and Suggestions for Authors

The authors addressed all my comments. 

Comments on the Quality of English Language

Minor editing is required. 
